# A Dual-Fusion Cognitive Diagnosis Framework for Open Student Learning Environments

## Abstract

Cognitive diagnosis model (CDM) is a fundamental and upstream component in intelligent education. It aims to infer students' mastery levels based on historical response logs. However, existing CDMs usually follow the ID-based embedding paradigm, which could often diminish the effectiveness of CDMs in open student learning environments. This is mainly because they can hardly directly infer new students' mastery levels or utilize new exercises or knowledge without retraining. Textual semantic information, due to its unified feature space and easy accessibility, can help alleviate this issue. Unfortunately, directly incorporating semantic information may not benefit CDMs, since it does not capture response-relevant features and thus discards the individual characteristics of each student. To this end, this paper proposes a dual-fusion cognitive diagnosis framework (DFCD) to address the challenge of aligning two different modalities, i.e., textual semantic features and response-relevant features. Specifically, in DFCD, we first propose the exercise-refiner and concept-refiner to make the exercises and knowledge concepts more coherent and reasonable via large language models. Then, DFCD encodes the refined features using text embedding models to obtain the semantic information. For response-related features, we propose a novel response matrix to fully incorporate the information within the response logs. Finally, DFCD designs a dual-fusion module to merge the two modal features. The ultimate representations possess the capability of inference in open student learning environments and can be also plugged in existing CDMs. Extensive experiments across real-world datasets show that DFCD achieves superior performance by integrating different modalities and strong adaptability in open student learning environments.

## 1 Introduction

Nowadays, intelligent education is gaining increasing attention in the field of computer science Liu (2021); Chen et al. (2023); Liu et al. (2023); Zhou et al. (2024). Cognitive diagnosis (CD), which is a fundamental upstream task in intelligent education Anderson et al. (2014), acts as a pivotal role in current student learning environments Liu (2021). It has a significant and primary impact on subsequent components such as computer adaptive testing Zhuang et al. (2022), course recommendations Huang et al. (2019); Xu & Zhou (2020), and learning path recommendations Liu et al. (2019). As illustrated in the left part of Figure 1, its goal is to deduce students' mastery level on each concept and other attributes, such as the difficulty levels of exercises through historical response logs and a Q-matrix.

Classical educational measurement cognitive diagnosis models (CDMs), such as item response theory (IRT) and the deterministic input, noisy and gate model (DINA) De La Torre (2009), either rely on hand-crafted interaction functions or stringent assumptions (e.g., students must master all concepts associated with an exercise to answer it correctly) or complex parameter estimation methods. These make them unsuitable for large-scale student learning environments. Consequently, neural-based CDMs have recently emerged rapidly. Most existing neural-based CDMs Wang et al. (2020a); Gao et al. (2021); Ma et al. (2022); Wang et al. (2023) follow the traditional ID-based embedding paradigm, vectorizing students, exercises and concepts through embeddings and distinguishing them by IDs. They subsequently update the ID-embeddings by recovering historical response logs (i.e., predict student score on exercises) through binary cross entropy (BCE) loss. However, adhering to this paradigm can lead to failure in open student learning environments where the number or content of students, exercises and concepts are dynamically changing. Students today often complete

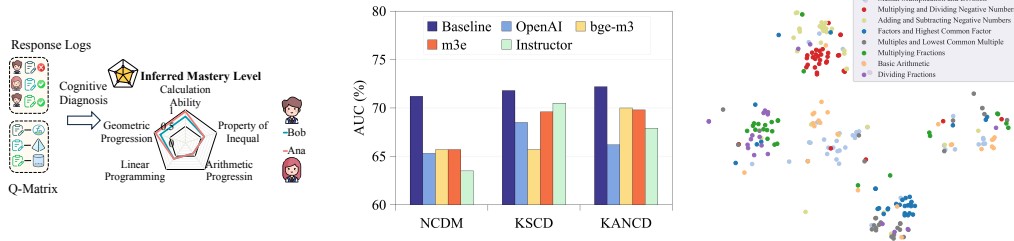

Figure 1: The left subfigure denotes the process of CD. The middle subfigure shows the results of the motivation study on MOOC-Radar dataset. The right subfigure shows the t-SNE visualization of exercise text via text-embedding-ada-002 from the NeurIPS2020 dataset, with each exercise point colored according to its corresponding concept. Notably, we select the subfigures of certain datasets for brevity. Similar results for other datasets are presented in the Appendix B.

tests on online education platforms such as IELTS, TOEFL, and GMAT. New students with a large number of their own response records can join at any time, and the assessment content may vary widely. And the online system must quickly diagnose the abilities of these new students and select subsequent test questions accordingly. Such a dynamic open student learning environment presents a significant drawback for the traditional ID-based CDM framework which relies on retraining to accommodate new students, exercises or concepts, because the extensive time required for retraining is often unacceptable given the low-latency demands of real-time testing. ***Therefore, our core idea is to design a framework that enables existing CDMs to be effective in open student learning environments without the need of retraining.***

Textual features (e.g., exercise text and concept name) have demonstrated the ability of generalizing to various downstream tasks in natural language processing due to their unified nature, even in unseen domains Radford et al. (2018; 2019); Brown et al. (2020). Clearly, textual features can potentially alleviate the aforementioned issue. All we need is to train a projector to map the textual space to the actual diagnostic space. However, to the best of our knowledge, textual CD is still unexplored. Unfortunately, as shown in the middle part of Figure 1, directly incorporating text semantic features in the traditional CD setting or open student learning environment may not benefit CDMs and can even perform worse than the original CDM. Details of this experiment can be found in Appendix B. We contend that two reasons account for this. First, as shown in the right part of Figure 1, exercises with the same concept are not well-clustered together and are even quite dispersed. It indicates that exercise text features may not directly reflect their related concepts. Second, as shown in Figure 5 of Appendix B, exercises with similar correctness rates are far apart. It indicates that textual features do not capture response-relevant features, thus disregarding the individual characteristics of each student. That is to say, simply incorporating textual information is not sufficient. We must also integrate other types of features, such as response-relevant features, to ensure the completeness of the diagnostic information.

To this end, this paper proposes a dual-fusion cognitive diagnosis framework (DFCD) to address the challenges of aligning two different modalities, namely, textual semantic features and response-relevant features. DFCD enables existing CDMs to be effective in open student learning environments without the need of retraining. Specifically, in DFCD, we first propose the exercise-refiner and concept-refiner to make the exercises and concepts more coherent and reasonable via large language models. Then, DFCD encodes the refined features using cutting-edge text embedding models to obtain the textual semantic features. For response-relevant features, we propose a novel response matrix to fully incorporate the information within the response logs and Q-Matrix, effectively balancing the size of feature spaces of students, exercises and concepts. Finally, DFCD designs a dual-fusion module to merge the two modal features. The ultimate representations possess the capability of inference in open student learning environments and can be also plugged in existing CDMs. Extensive experiments across real-world datasets show that DFCD achieves superior performance by integrating representations in different modalities and strong adaptability in open student learning environments.

The subsequent sections respectively recap the related work, present the preliminaries, introduce the proposed DFCD, show the empirical analysis and finally conclude the paper.

## 2 RELATED WORK

### 2.1 COGNITIVE DIAGNOSIS MODELS

**ID-based Cognitive Diagnosis Models.** Most existing CDMs adhere to the ID-based embedding paradigm, which involves vectorizing students, exercises, and concepts through embeddings and distinguish them by their IDs. They can be categorized by the dimension of mastery levels into two types: latent factor models (e.g., using a fixed length vector to represent students' latent mastery levels), such as multidimensional item response theory (MIRT)Sympson (1978), and models based on patterns of concept mastery (i.e., the dimension of mastery level is the number of concepts), such as DINA De La Torre (2009). These two methods either rely on hand-crafted interaction functions or impose stringent assumptions and complex parameter estimation methods, which may not be effective in today's large-scale student learning environments. NCDM Wang et al. (2020a) employs multi-layer perceptrons (MLP) as interaction function and represents mastery patterns as continuous variables within the range of $[0, 1]$. Various approaches have been employed to capture fruitful information in the response logs, such as MLP-based Ma et al. (2022); Wang et al. (2023), graph attention network based Gao et al. (2021), Bayesian network based Li et al. (2022). However, this paradigm can fail in open student learning environments. Due to the limitations of IDs, for instance, ID-embedding methods require model retraining for new students, which is unacceptable in real online platforms where timely diagnostic results are expected.

**Cognitive Diagnosis Models for Open Student Learning Environments.** As online education platforms become increasingly popular, designing CDMs for open student learning environments is crucial. ICD Tong et al. (2022) makes the first attempt to target streaming log data with the goal of updating students' mastery levels in real-time without the need for retraining. However, it may require substantial time when there are numerous records in a short period. DCD Chen et al. (2023), IDCD Li et al. (2024) and ICDM Liu et al. (2024a) rely on simple interaction matrices or hand-crafted graph structures as the feature space, which either demonstrate unpromising performance in open student learning environments or solely focus on a single scenario (e.g., new students). And it is worth noting that unlike the cold-start issues addressed by TechCD Gao et al. (2023) and ZeroCD Gao et al. (2024), open student learning environment focus on inferring the attribue for new students, new exercises and new concepts with unseen response logs during the training phase, which is commonly seen in current online education or testing platforms.

### 2.2 TEXT-BASED REPRESENTATION LEARNING IN INTELLIGENT EDUCATION SYSTEMS

Text-based representation learning in intelligent education systems has recently gained significant popularity. NCDM+ Wang et al. (2020a) utilizes exercise text via TextCNN Kim (2014) to complete the Q-Matrix in CD. EKT Liu et al. (2021) enhances student performance prediction in knowledge tracing by utilizing exercise text descriptions. However, neither of them fuse the exercise text or concept name into representations in CD. The most related work is ECD Zhou et al. (2021), which fuses student context-aware features (e.g., parental education level, monthly study expenses) into representations of students in cognitive diagnosis. However, such features are often difficult to obtain in real-world scenarios due to the need to protect the privacy of students and teachers. TechCD Gao et al. (2023) and ZeroCD Gao et al. (2024) use BERT Devlin (2018) for simply extracting exercise text feature which is different from our focus.

## 3 PRELIMINARIES

Let us consider open student learning environments which contain three sets: $S = \{s_1, \ldots, \}, E = \{e_1, \ldots, \}$, and $C = \{c_1, \ldots, \}$. The relationship between exercises and concepts is represented by the matrix $\mathbf{Q}$, which is a binary matrix where $\mathbf{Q}_{jk} = 1$ denotes exercise $e_j$ is related to concept $c_k$. In this paper, we consider three types of open learning environments: unseen students, unseen exercises, and unseen concepts. For instance, in the unseen students scenario, the number of exercises and concepts **remains unchanged**. Notably, this means we do not consider overlapping open scenarios, such as the simultaneous occurrence of a large number of new students and new exercises. This is because data from online learning platforms can always be divided into the aforementioned three types of open learning environments based on timestamps.

Figure 2: The overall framework of DFCD. (a) Textual feature constructor. Examples in it are all from real data. (b) Response feature constructor. (c) Detailed components of DFCD.

**Problem Definition.** Suppose that the open learning student environment has collected a large number of observed response logs, represented as triplets $T^O = \{(s, e, r) | s \in S^O, e \in E^O, r_{se} \in \{0, 1\}\}$. $r_{se} = 1$ represents correct and $r_{se} = 0$ represents wrong. $S^O$ denotes the observed student set in $T^O$, and similarly, $E^O$ and $C^O$ represent the observed sets of exercises and concepts, respectively. Assume that there are a certain number of unobserved upcoming response logs $T^U$ involving $S^U$, $E^U$ and $C^U$. The goal of CD in open student learning environment is to infer the $\mathbf{Mas} \in \mathbb{R}^{|S^U| \times |C^O \cup C^U|}$ which denotes the latent mastery level of students on each concept.

## 4 METHODOLOGY: THE PROPOSED DFCD

In this section, we present the textual feature constructor and response feature constructor. Following that, we delve into the proposed dual-fusion framework. We conclude the section by discussing the model's training. Notably, the strength of DFCD lies in addressing CD in open learning environments. Hence, all its underlying notions are derived from this scenario. Nevertheless, we assert that DFCD is versatile enough to be applied in standard scenarios like previous works Wang et al. (2020a). The framework of DFCD is shown in Figure 2.

### 4.1 TEXTUAL FEATURE CONSTRUCTOR

The exercise text can, to some extent, reflect the difficulty level of specific concepts for the students. However, it is evident that exercise text alone cannot directly reflect the expert annotated concepts being tested. For instance, as shown in Figure 2(a), it may related to many concepts (e,g, trigonometric functions, calculate ability), but the annotated concept is "Square Roots". The name of the concept also has this issue; the same concept, such as "time" is completely different in physics and mathematics. To bridge the gap between real text and its inherent concepts, inspired by the recent successes of large language models (LLMs) in reasoning, we utilize LLMs as exercise refiner and concept refiner. Specifically inspired by recent advancements Xi et al. (2023); Ren et al. (2024), we design the system prompt $\alpha_e, \alpha_c$ to function as part of the input for LLMs. This prompt aims to explicitly outline the LLM's role in creating precise summarizations for exercises or concepts by clearly defining the input-output content and the desired output format. By combining this system prompt with the exercise/concept summarization generation prompts $\beta_e$ and $\beta_c$, we can effectively harness LLMs to create precise summarizations. We provide vivid examples in Appendix C. The mathematical process is as follows:

$$\mathcal{S}_{e_j} = \mathbf{LLM}(\alpha_e, \beta_e, \gamma_{e_j}), \quad \mathcal{S}_{c_k} = \mathbf{LLM}(\alpha_c, \beta_c, \gamma_{c_k}), \quad (1)$$

where $\mathcal{S}_{e_j}$ denotes the summarization result of $e_j$, $\mathcal{S}_{c_k}$ denotes the summarization result of $c_k$. $\gamma_{e_j}$ represents the related concept name of $e_j$, $\gamma_{c_k}$ represents exercises which assess $c_k$. Finally, we can obtain the refined textual features of exercises and concepts using advanced text embedding models.

These models effectively transform diverse text inputs into fixed-length vectors, preserving their inherent meaning and contextual information. It can be expressed as

$$\mathbf{Z}_{e_j}^{(1)} = \textbf{TEM}(\mathcal{S}_{e_j}), \quad \mathbf{Z}_{c_k}^{(1)} = \textbf{TEM}(\mathcal{S}_{c_k}), \tag{2}$$

where $\mathbf{Z}_{e_j}^{(1)} \in \mathbb{R}^{1 \times d_l}$ denotes the refined textual feature of exercise $e_j$, $\mathbf{Z}_{c_k}^{(1)} \in \mathbb{R}^{1 \times d_l}$ denotes the refined textual feature of concept $c_k$. **TEM** denotes any text embedding modules (e.g., text-embedding-ada-002 Brown et al. (2020), instructor Su et al. (2022)). $d_l$ is the dimension of text embedding in **TEM**. Notably, since student textual profiles are difficult to obtain due to privacy and educational sensitivity, we derive student textual features $\mathbf{Z}_{s_i}^{(1)}$ as the pooled (e.g., mean) result of the exercises they have completed. We provide the t-SNE visualization of text embeddings before and after refinement in the Appendix C, where it can be observed that most exercises with the same concepts are clustered more together than before refinement.

## 4.2 RESPONSE FEATURE CONSTRUCTOR

As shown in Figure 1, we contend that directly replacing the ID-embedding with text embedding fails primarily because the textual descriptions do not accurately reflect the actual context of student responses. For instance, a question might have a simple textual description, which could result in an embedding that reflects a lower difficulty level. However, certain details may be prone to errors, significantly reducing the students' accuracy and revealing a higher actual difficulty level. Therefore, fusing response feature into the representations is also very crucial. The previous work Chen et al. (2023); Li et al. (2024), following the paradigm of recommendation systems Liang et al. (2018), utilizes the historical interaction matrix $\mathbf{I}^O$ as features for students or exercises. This approach may lead to an imbalance in the size of the student and exercise feature space, causing it to fail in certain open student learning environments, and fails to incorporate characteristics of the concepts, which have shown success in recent works Ma et al. (2022); Wang et al. (2023). To this end, we propose the response matrix $\mathbf{R}^O \in \mathbb{R}^{(|S^O|+|E^O|+|C^O|) \times (|S^O|+|E^O|+|C^O|)}$ which incorporate both $\mathbf{I}^O$ and $\mathbf{Q}^O$ and balance the size of feature space well. It can be elegantly expressed in matrix form

$$\mathbf{R}^O = \begin{pmatrix} \mathbf{O} & \mathbf{I}^O & \mathbf{O} \\ \mathbf{I}^{O^\top} & \mathbf{O} & \mathbf{Q}^O \\ \mathbf{O} & \mathbf{Q}^{O^\top} & \mathbf{O} \end{pmatrix}, \mathbf{Z}_{s_i}^{(2)} = \mathbf{R}_{s_i}^O, \mathbf{Z}_{e_j}^{(2)} = \mathbf{R}_{e_j+|S^O|}^O, \mathbf{Z}_{c_k}^{(2)} = \mathbf{R}_{c_k+|S^O|+|E^O|}^O. \tag{3}$$

As shown in equation 3, students' features consist of their responses to exercises, exercises' features consist of student responses and their related concepts, and concepts' features consist of the exercises that assess them. We can easily derive the response features from $\mathbf{R}^O$ as shown in the right part of equation 3, namely, $\mathbf{Z}_{s_i}^{(2)}$, $\mathbf{Z}_{e_j}^{(2)}$ and $\mathbf{Z}_{c_k}^{(2)} \in \mathbb{R}^{1 \times (|S^O|+|E^O|+|C^O|)}$.

## 4.3 DUAL FUSION FRAMEWORK

**Projectors.** After obtaining the textual features and response features, the key challenge is how to fuse these two modalities, which have different dimensions, in a personalized manner. Firstly, we introduce T-Projector and R-Projector to align features from two modalities in the same dimension, facilitating subsequent processing. Concretely, in each projector, we utilize three different MLP for students, exercises, and concepts. Here, we take student $s_i$ as an example. It can be expressed as

$$\tilde{\mathbf{Z}}_{s_i}^{(1)} = \text{MLP}_s^{(1)}(\mathbf{Z}_{s_i}^{(1)}), \quad \tilde{\mathbf{Z}}_{s_i}^{(2)} = \text{MLP}_s^{(2)}(\mathbf{Z}_{s_i}^{(2)}), \tag{4}$$

where $\tilde{\mathbf{Z}}_{s_i}^{(1)}, \tilde{\mathbf{Z}}_{s_i}^{(2)} \in \mathbb{R}^{1 \times d}$ denotes the aligned student features in the dual modalities. $\text{MLP}_s^{(1)}$ and $\text{MLP}_s^{(2)}$ are trainable neural networks to change the dimension into $d$.

**Personalized Attention Module.** As our goal is to infer the mastery level of students, which is determined by the aforementioned two modalities, each student should have different weights assigned to these modalities. This reflects the personalized nature of student learning in reality. Therefore, inspired by Wang et al. (2021); Liu et al. (2024a), we design a personalized attention module. The weight corresponding to the two modality can be computed as

$$w_{s_i}^{(1)} = \mathbf{a}_s \tanh\left(\tilde{\mathbf{Z}}_{s_i}^{(1)} \mathbf{W}_s + \mathbf{b}_s\right)^\top, \, w_{s_i}^{(2)} = \mathbf{a}_s \tanh\left(\tilde{\mathbf{Z}}_{s_i}^{(2)} \mathbf{W}_s + \mathbf{b}_s\right)^\top, \tag{5}$$

where $\mathbf{a}_s \in \mathbb{R}^{1 \times d}$ denotes attention vector, $\mathbf{W}_s^g \in \mathbb{R}^{d \times d}$ and $\mathbf{b}_s^g \in \mathbb{R}^{1 \times d}$ are trainable parameters in the students' features fusion phase. We can derive the ultimate representation of $s_i$ by normalized weighted summed of $\tilde{\mathbf{Z}}_{s_i}^{(1)}$ and $\tilde{\mathbf{Z}}_{s_i}^{(2)}$ which can be expressed as

$$\tilde{w}_{s_i}^{(1)} = (1 + e^{w_{s_i}^{(2)} - w_{s_i}^{(1)}})^{-1}, \quad \tilde{w}_{s_i}^{(2)} = (1 + e^{w_{s_i}^{(1)} - w_{s_i}^{(2)}})^{-1}, \quad \mathbf{Z}_{s_i} = \tilde{w}_{s_i}^{(1)} \tilde{\mathbf{Z}}_{s_i}^{(1)} + \tilde{w}^{(2)} \tilde{\mathbf{Z}}_{s_i}^{(2)}, \quad (6)$$

where $\tilde{w}_{s_i}^{(1)}$ and $\tilde{w}_{s_i}^{(2)}$ denotes the normalized weights. $\mathbf{Z}_{s_i}$ represents the fused representation of student $s_i$. Similarly, one can obtain $\mathbf{Z}_{e_j}$ and $\mathbf{Z}_{c_k}$ through the same process.

**Graph Encoder.** Previous works Gao et al. (2021); Liu et al. (2024a) have shown that extracting the relationships among students, exercises, and concepts is crucial, as it can enhance the model's generalization and interpretability performance. Therefore, we utilize a cutting-edge graph encoder to obtain the final representation of $s_i$, which can be expressed as $\mathbf{H} = \mathbf{Encoder}(\mathbf{Z}_s, \mathbf{Z}_e, \mathbf{Z}_c)$ where **Encoder** can be any graph encoder like graph attention network Brody et al. (2022) or graph transformer Shi et al. (2021). Details can found in Appendix C.

## 4.4 TRAINING FOR DFCD

**Integrating Existing CDMs.** To integrate DFCD with most existing CDMs, we need to modify the dimensions to align with the specific type of CDM being used. Since our goal is to infer the students' mastery levels in a fixed dimension, we assume that the total number of concepts is already known (i.e., $|C^O| + |C^U|$). For CDMs where the embedding size is a latent dimension (e.g., KaNCD), we directly employ $\mathbf{H}_{s_i}, \mathbf{H}_{e_j}$ and $\mathbf{H}_{c_k}$ as the input embedding for the integrated CDMs. Otherwise (e.g., NCDM), following Liu et al. (2024a), we introduce transformation layers. Here, we take student $s_i$ as an example, which can be formulated as

$$\tilde{\mathbf{H}}_{s_i} = \mathbf{H}_{s_i} \mathbf{W}_{\mathrm{t}}^{(s)} + \mathbf{b}_{\mathrm{t}}^{(s)}, \tag{7}$$

where $\tilde{\mathbf{H}}_{s_i}$ will be employed as input embedding for incorporated CDMs and $\mathbf{W}_{\mathrm{t}}^{(s)} \in \mathbb{R}^{d \times (|C^O| + |C^U|)}$, $\mathbf{b}_{\mathrm{t}}^{(s)} \in \mathbb{R}^{1 \times (|C^O| + |C^U|)}$ are trainable parameters. This significantly reduces the time complexity of graph convolution by encoder which will be further analyzed in the Appendix C.4. Therefore, we train the DFCD with integrated CDMs in an end-to-end manner.

**SimpleCD.** Existing neural-based CDMs Gao et al. (2021); Wang et al. (2023); Liu et al. (2024a) except NCDM often have numerous parameters, which may not be effective in open learning environments because they tend to overfit the historical response logs Li et al. (2024). Therefore, we propose a CDM called "SimpleCD" which is **parameter-free** except for the interaction function. It can be expressed as

$$\hat{y}_{ij} = \mathcal{F}\left((\sigma(\mathbf{H}_{s_i} \mathbf{H}_c^\top) - \sigma(\mathbf{H}_{e_j} \mathbf{H}_c^\top)) \odot \mathbf{Q}_{e_j}\right), \tag{8}$$

where $\hat{y}_{ij} \in [0, 1]$ represents the prediction score of $i$-th student practice $j$-th exercise, $\mathcal{F}(\cdot)$ denotes the Positive MLP which is commonly utilized in CD and $\sigma$ typically employs the Sigmoid. $\sigma(\mathbf{H}_{s_i} \mathbf{H}_c^\top \in \mathbb{R}^{1 \times (|C^O| + |C^U|)})$ denotes the mastery level of student $s_i$, namely $\mathbf{Mas}_{s_i}$. "$\odot$" represents the element-wise product. $\mathbf{Q}_{e_j} \in \mathbb{R}^{1 \times (|C^O| + |C^U|)}$ signifies the concepts associated with the $j$-th exercise. More details about Postive MLP and SimpleCD can be found in Appendix C. We empirically find that it works well in open student learning environments.

**Optimization.** Given input features of students, exercises and concepts, existing CDMs can predict the score of students on certain exercises, which can be formulated as

$$\hat{y}_{ij} = \mathcal{M}_{\mathrm{CD}}(\mathbf{H}_{s_i}, \mathbf{H}_{e_j}, \mathbf{H}_c), \tag{9}$$

where $\mathcal{M}_{\mathrm{CD}}(\cdot)$ denotes the CDMs, and $\mathbf{H}$ represents the input features that contains the representation of the student, exercises and concepts. In the CD task, the main loss function involves computing the BCE loss between the actual response scores and the model's predicted outcomes in a mini-batch. This overall loss can be expressed as follows

$$\mathcal{L}_{\mathrm{BCE}} = -\sum_{(s,e,r_{se}) \in T^O} [r_{se} \log \hat{y}_{se} + (1 - r_{se}) \log(1 - \hat{y}_{se})]. \tag{10}$$

**Training Cost.** We have conducted a complexity analysis and training speed comparison in Appendix C.4. Notably, after training, we can infer the mastery level of 126 newly arrived students

with 1,024 response logs in just 64 ms. For the cost of using large language models, our strategy for selecting large language models is discussed in Appendix D.4. We found that using cost-effective models like OpenAI's GPT-3.5-Turbo or Google's Gemini-pro achieves relatively satisfactory results.

## 5 EXPERIMENT

In this section, we first delineate three real-world datasets and evaluation metrics. Then through comprehensive experiments, we aim to manifest the preeminence of DFCD in both open student learning environment and standard scenario. **Due to space constraints, we place the experiments in the standard scenario in Appendix D.5**. To ensure reproducibility and robustness, all experiments are conducted ten times. Our code is available at `https://anonymous.4open.science/r/DFCD-8710`.

### 5.1 EXPERIMENTAL SETTINGS

**Datasets.** Our experiments are conducted on three real-world datasets, i.e., NeurIPS2020 Wang et al. (2020b), XES3G5M Liu et al. (2024b) and MOOCRadar Yu et al. (2023). These three datasets represents diverse educational contexts and subject, which are collected from a wide variety of courses includes the educational contexts and subjects from chinese, history, economics ,math, physics and so on. For more detailed statistics on these three datasets, please refer to Table 1. The details about datasets source and data preprocessing are depicted in the Appendix D.1. Notably, "Sparsity" refers to the sparsity of the dataset, which is calculated as $\frac{|T|}{|S||E|}$. "Average Correct Rate" represents the average score of students on exercises, and "**Q** Density" indicates the average number of concepts per exercise.

Table 1: Statistics of real-world datasets for experiments.

| Datasets | #Students | #Exercises | #Concepts | #Response Logs | Sparsity | Average Correct Rate | **Q** Density |
|---|---|---|---|---|---|---|---|
| NeurIPS2020 | 2,000 | 454 | 38 | 258,233 | 0.284 | 0.547 | 1.000 |
| XES3G5M | 2,000 | 1,624 | 241 | 207,204 | 0.063 | 0.817 | 1.000 |
| MOOCRadar | 2,000 | 915 | 696 | 385,323 | 0.210 | 0.878 | 2.240 |

**Evaluation Metrics.** To assess the efficacy of DFCD, we utilize both score prediction and interpretability metrics following the previous works Wang et al. (2020a); Chen et al. (2023). This approach offers a holistic evaluation from both the predictive accuracy and interpretability standpoints.

Score Prediction Metrics: Evaluating the efficacy of CDMs poses difficulties owing to the absence of the true mastery level. A prevalent workaround is to appraise these models based on their capability to predict students' scores on exercises in the test data. The classic classification metrics such as area under the curve (AUC), Accuracy (ACC) are used in our paper.

Interpretability Metric: Diagnostic results are highly interpretable hold significant importance in CD. In this regard, we employ the degree of agreement (DOA), which is consistent with the approach used in Wang et al. (2020a); Li et al. (2022). The detailed description about DOA can be found in Appendix D.2. We compute the top 10 concepts with the highest number of response logs in our experiment and refer to it as DOA@10.

**Implementation Details.** For parameter initialization, we employ the Xavier Glorot & Bengio (2010), and for optimization purposes, Adam Kingma & Ba (2015) is adopted. The batch size is set as 1024 for all datasets. The learning rate is fixed as $1e^{-4}$. We adjust the dimension $d$ within the range $\{32, 64, 128, 256\}$, the type of graph encoder within the range $\{MLP, GCN, GAT, GT\}$. We utilize four attention heads for attention-based encoders, with all other parameters set to the PyG Fey & Lenssen (2019) defaults. We employ grid search to find the best hyperparameters using the validation set. Selection related to LLMs is introduced in Appendix D.4. Analysis regarding the aforementioned hyperparameters can be found in Section 5.3 and Appendix D.9.

### 5.2 PERFORMANCE COMPARISON IN OPEN STUDENT LEARNING ENVIRONMENT

**Compared Methods.** We compare DFCD against other methods and utilize the hyperparameter settings described in their respective original publications. More details can be found in Appendix D.

Table 2: Overall performance in open student learning environment scenario. In each column, an entry with the best mean value is marked in bold and underline for the runner-up. The standard deviation is not shown in the table since it is very small (less than 0.01). If the mean value of the best model significantly differs from the runner-up, passing a $t$-test with a significance level of 0.05, then we denote it with "*" at the corresponding position. "-" indicates that the model is not suitable of calculating this metric.

| Dataset | NeurIPS2020 | | | XES3G5M | | | MOOCRadar | | |
|---|---|---|---|---|---|---|---|---|---|
| Metric | AUC | ACC | DOA@10 | AUC | ACC | DOA@10 | AUC | ACC | DOA@10 |
| Unseen Student | | | | | | | | | |
| KANCD-Mean | 66.60 | 62.18 | - | 71.23 | 82.32 | - | 81.60 | 88.70 | - |
| KANCD-Nearest | 74.59 | 68.00 | 71.15 | 71.55 | 81.97 | 60.27 | 89.37 | 90.34 | 77.98 |
| IDCD | 77.64 | 70.65 | 74.15 | 75.68 | 82.29 | 69.75 | 92.36 | 91.32 | 81.26 |
| ICDM | 67.67 | 62.99 | 62.53 | 70.34 | 81.53 | 61.82 | 86.94 | 89.23 | 71.10 |
| **DFCD** | 78.19 | 71.39* | 74.33 | 77.79* | 83.05 | 71.99* | 92.91 | 91.68 | 82.15 |
| Unseen Exercise | | | | | | | | | |
| KANCD-Mean | 67.61 | 62.86 | 70.49 | 55.68 | 77.60 | 58.63 | 59.60 | 62.03 | 74.17 |
| KANCD-Nearest | 69.58 | 69.12 | 70.01 | 55.34 | 74.12 | 58.58 | 65.14 | 69.85 | 75.59 |
| IDCD | 74.63 | 68.28 | 73.90 | 62.30 | 77.27 | 67.09 | 78.52 | 87.79 | 81.07 |
| ICDM | 69.49 | 64.17 | 64.80 | 61.10 | 79.03 | 63.18 | 79.79 | 87.06 | 73.71 |
| **DFCD** | 77.76* | 71.29* | 74.17* | 76.15* | 82.61* | 71.82* | 91.98* | 91.61* | 81.93 |
| Unseen Concept | | | | | | | | | |
| KANCD-Mean | 67.91 | 65.61 | 68.21 | 63.01 | 71.57 | 58.89 | 82.30 | 85.58 | 76.48 |
| KANCD-Nearest | 70.53 | 65.80 | 68.53 | 65.38 | 81.67 | 57.95 | 84.69 | 87.22 | 76.44 |
| IDCD | 73.55 | 66.36 | 68.04 | 72.50 | 82.04 | 69.51 | 91.12 | 91.01 | 81.27 |
| ICDM | 73.43 | 66.40 | 61.08 | 70.75 | 82.04 | 61.53 | 92.15 | 91.18 | 68.08 |
| **DFCD** | 77.68* | 70.68* | 73.85* | 78.83* | 83.41* | 72.14* | 92.89* | 91.56* | 82.10 |

• KaNCD-Mean Wang et al. (2023): As the original KaNCD is designed solely for the standard scenario. We assigns the embedding of unseen students or exercises to the average of the seen ones Liu et al. (2024a).

• KaNCD-Nearest Wang et al. (2023): For each unseen students, exercises or concepts in $T^U$, we assign their embedding based on the most similar one in $T^O$, who is selected based on the similarity of response logs. Here, we use cosine similarity as the similarity measure function Liu et al. (2024a).

• IDCD Li et al. (2024): It propose an identifiable cognitive diagnosis framework based on a novel response-proficiency response paradigm and its diagnostic module leverages inductive learning representations which can be used in the open student learning environment.

• ICDM Liu et al. (2024a): It utilizes a student-centered graph and inductive mastery levels as the aggregated outcomes of students' neighbors in student-centered graph which enables to infer the unseen students by finding the most suitable representations for different node types.

**Details.** To evaluate the effectiveness of our proposed DFCD in open student learning environments, we conduct experiments following Liu et al. (2024a) on datasets with unseen students, unseen exercises, and unseen concepts. For the unseen student scenario, we randomly select students who do not appear in the training data. For the unseen exercise scenario, we randomly select exercises not present in the training data. For the unseen concept scenario, we randomly select exercises with concepts that are not in the training data. The test size $p_t$ is set to 0.2, following the previous researches Wang et al. (2020a); Li et al. (2022). In order to prevent data leakage, we retain the test data intact and partition the training data by students, exercises, or concepts at a ratio of 0.2, with the validation ratio set at 0.1. In this approach, we can obtain two sets from training data: $T^O$ and $T^U$. We train the DFCD using only the $T^O$. Then we use the $T^U$ for inference. *Ultimately, the score prediction metrics is computed only by the prediction of students set $S^U$ in $T^U$ for exercises in the test data.* We provide an example of how our DFCD trains and infers in the open student learning environment scenario in Appendix D.3. KaNCD-Mean which assigns the embedding of unseen students to the average of the seen ones during the training process has the same representation on every students in test set. So it is not suitable for calculating DOA. In Table 2, we use "-" to indicate this inapplicability.

**Results.** The comparison results are listed in Table 2. We have the following key observations:

• ID-Based CDMs with a simple postprocessing such as the strategy of mean or finding the nearest representation may solve the problem of open student learning environment to some extent. However, they still don't produce satisfactory results and fall significantly short compared to the outcomes of other models. For IDCD and ICDM, which is specifically designed for open student learning environment, they perform better than the standard CDMs in most of the cases,

• DFCD consistently outperforms the other models on all datasets and scenario. This demonstrates that DFCD is more effective in the open student learning environment scenario in CD. And it is worth mentioning that DFCD has such a great performance gap between other models especially in the unseen exercise and knowledge scenario, this may be because the CD designed for open student learning environment like IDCD and ICDM focus mainly on the unseen student. Due to the fusion of textual features and response-relevant features, DFCD has a strong adaptability and interpretability in all scenarios of the open student learning environments.

**Ablation Study.** To showcase the contributions of each component in DFCD, we conduct an ablation study on DFCD, which is divided into the following three versions: DFCD-w.o.TE: This version removes the text semantic embeddings. DFCD-w.o.RE: This version removes the response-relevant embeddings. DFCD-w.o.attn: This version removes the attention module when fuse the text semantic embeddings and response-relevant embeddings, the fusion ratio is simply set to 0.5 on both embeddings. As shown in Table 6, DFCD surpasses almost all the versions in both prediction and interpretability performance. This suggests that these components, when combined, enhance DFCD. When each component is removed individually, either the prediction performance decreases or the interpretability performance suffers, indicating that textual features and response-relevant features is both important for the performance of the DFCD and the fusion method of these two representations is also crucial. The DOA@10 on MOOC-Radar is higher in all scenarios when removing the response-relevant features. This may be because there are 696 concepts. To align with previous methods, we select DOA@10, but it may not adequately represent all concepts.

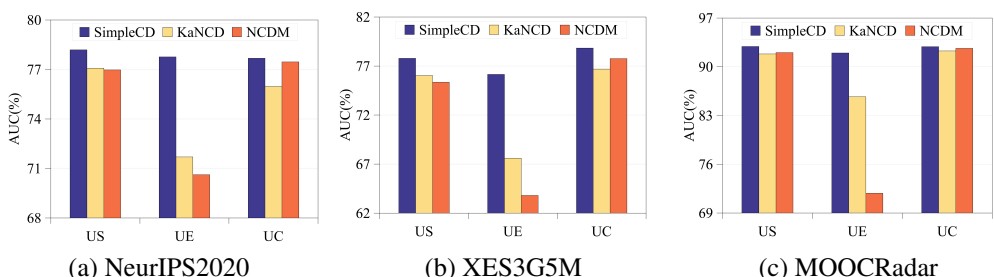

(a) NeurIPS2020     (b) XES3G5M     (c) MOOCRadar

Figure 3: Comparison of DFCD with different integrated CDMs. US means the scenario of unseen student, UE means the scenario of unseen exercise, and UC means the scenario of unseen concept.

**Versatility Analysis.** To showcase the versatility of DFCD, we incorporate the fused features generated by DFCD into the commonly used CDM. In this experiment, we compare our proposed SimpleCD with NCDM Wang et al. (2020a) and KaNCD Wang et al. (2023). For brevity, we abbreviate unseen students as US, unseen exercises as UE, and unseen concepts as UC. As shown in Figure 3, the proposed SimpleCD demonstrates superior performance in open student learning environment compared. This improvement might be attributed to the overly simplistic interaction function in NCDM, which may falls due to the weak knowledge problem Wang et al. (2023), resulting in limited information acquisition. In open student learning environment with inherently scarce data, this leads to significantly poor performance. While KaNCD suffers from excessive parameters, which may makes it overfitting to historical response logs much more seriously Li et al. (2024). The less parameters and the ability on effective information acquisition of SimpleCD contribute to the higher performance in open student learning environment scenario.

**Generalization Analysis.** To assess the efficacy of DFCD's generalization ability, we conduct experiments on three datasets with varying test size $p_t = \{0.1, 0.2, 0.3, 0.4, 0.5\}$. As $p_t$ increases which is consistent with Gao et al. (2021), the generalization ability of CDMs is tested more stringently. As depicted in Figure 11, with an increasing $p_t$, the number of response logs used for training decreases. However, DFCD consistently outperforms IDCD and ICDM in the open student

learning environment scenario, indicating that DFCD can provide more accurate diagnosis results with fewer response logs. Moreover, DFCD decrease more slightly with the increasing $p_t$ than others. This is particularly suitable for current online education platform, where students often have limited response logs. And we also conduct the experiment on cold-start scenario where response logs per new students are sparse. We compare our DFCD with the SOTA model BetaCD Bi et al. (2023) and show a competitive result with it in Table 7.

**Diagnosis Result Analysis.** Indeed, students can naturally be grouped into categories based on their scores, such as those with low and high correct rates. This classification reflects intrinsic differences in their mastery levels. Details can be found in Appendix D.11. As shown in Figure 14, DFCD displays a long strip trend, with the color of the points on the strip gradually changing from lighter to darker shades. This indicates that DFCD successfully captures both the historical and new students' **Mas** trends. In contrast, the color distribution of IDCD is relatively loose, suggesting it may fail to accurately capture students' **Mas** information. Moreover, the mastery levels of new students inferred by DFCD are more reliable, as new students with similar correct rates (colored in green) cluster closely with historical students (colored in blue) of comparable rates.

## 5.3 Hyperparameter Analysis

**The Effect of Text Embedding Model.** As shown in Figure 13 in Appendix D.10, in most scenarios and datasets, text-embedding-ada-002 and bge-m3 demonstrate superior performance, likely due to their extensive training data, which supports them to better capture semantic information. Details can be found in Appendix D.10. Other hyperparameter analysis can be found in Appendix D.9.

**The Effect of Dimension $d$.** The dimension $d$ determine the dimension of the transformed text semantic embeddings and response-relevant embeddings. As shown in Figure 12(a)(b)(c), the performance achieve the highest point at 64 or 128 in most cases, so it is recommended to set the $d$ either 64 or 128 to achieve the best results in the model's performance.

**The Effect of Different Graph Encoder.** We evaluate the impact of different graph encoders on DFCD in Figure 12(d)(e)(f). Attention-based encoders (e.g., GAT, GT) outperform GCN, as open learning environments resemble the inductive setting in graph representation learning. While MLP achieves decent results due to our strong fused representation, but the addition of the graph structure can catch more information of the relation between students, exercise and concept and perform better in such a complex open student learning environment. GT generally excels as it considers all nodes, not just local neighborhoods like GAT, making it our recommended default encoder.

**The Effect of Mask Ratio.** The mask is used for graph encoder for the purpose of the robustness of models. As shown in Figure 12(g)(h)(i), there is an improvement when using the mask in the models. And the performance become stable after the threshold of 0.3. Based on these observations, it is advisable to set the mask ratio within the range of 0.2 to 0.3 to achieve optimal performance.

## 6 Conclusion And Discussion

**Conclusion.** This paper proposes an dual fusion cognitive diagnosis framework (DFCD), where most existing CDMs can be integrated. For the first time, we identify that directly utilizing exercise text features may not benefit CDMs and can even degrade their performance. Therefore, we leverage LLMs as refiners to enhance the textual content. Via DFCD, we fuse the textual features with response-relevant features and integrating existing CDMs to achieve remarkable performance in open student learning environments on three real-world datasets. Our work enables the CDM to better grasp the semantic meaning of exercise through leveraging LLMs' inference capabilities and provides a way to combine textual information and response information which allows CDM for a more comprehensive understanding of student performance by utilizing multiple data sources.

**Discussion.** In the future, we plan to incorporate additional textual features, such as students' family economic conditions or teacher quality, to further enhance the relevance and precision of the student profile. We also aim to explore more prompt combinations or introduce suitable fine-tuning techniques to help large language models filter out noise within the textual features, thereby reducing potential biases. Additionally, we plan to extend DFCD to be effective in other scenarios of intelligent education, making our model applicable to a wider range of cases.

## 7 STATEMENT

### 7.1 ETHICS STATEMENT

In this paper, we have adhered to the ethical guidelines outlined in the ICLR Code of Ethics `https://iclr.cc/public/CodeOfEthics`. Specifically, the research presented does not involve human subjects or raise concerns related to privacy, security, or legal compliance. The datasets used in this study are publicly available, and their use complies with all applicable licenses and terms of use.

### 7.2 REPRODUCIBILITY STATEMENT

We have taken several steps to ensure the reproducibility of the results presented in this paper. Detailed descriptions of datasets and implementation are provided in Sections 5.1 of the main paper. We also provide our data and code in the anonymous repository at `https://anonymous.4open.science/r/DFCD-8710`.

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

APPENDIX

## A    NOTATIONS

Table 3: The notions involved in this paper.

| Notation | Definition |
|---|---|
| CD | Cognitive Diagnosis. |
| CDMs | Cognitive Diagnosis Models. |
| **Mas** | The diagnostic result of CDMs, i.e., mastery levels of students. |
| $S^O$ | The set of observed students. |
| $E^O$ | The set of observed exercises. |
| $C^O$ | The set of observed concepts. |
| $S^U$ | The set of unobserved students. |
| $E^U$ | The set of unobserved exercises. |
| $C^U$ | The set of unobserved concepts. |
| $r_{se}$ | The ground truth score of student $s_i$ practice exercise $e_j$. |

## B    DETAILS ABOUT THE MOTIVATION STUDY

Here, we will provide some details about the motivation study.

**Details about middle subfigure in Figure 1.** We empirically find that directly replacing the ID-embeddings with text embeddings may not benefit CDMs and can even degrade their performance. In this study, we focus on vectorized the exercise text via cutting-edge text embedding modules. To demonstrate this, we conduct experiments on three widely used CDMs using four types of text embeddings, namely text-embedding-ada-002 Brown et al. (2020), BGE-M3 Chen et al. (2024), M3E-base Wang Yuxin (2023), and Instructor-base Su et al. (2022). Here, we utilize AUC as the

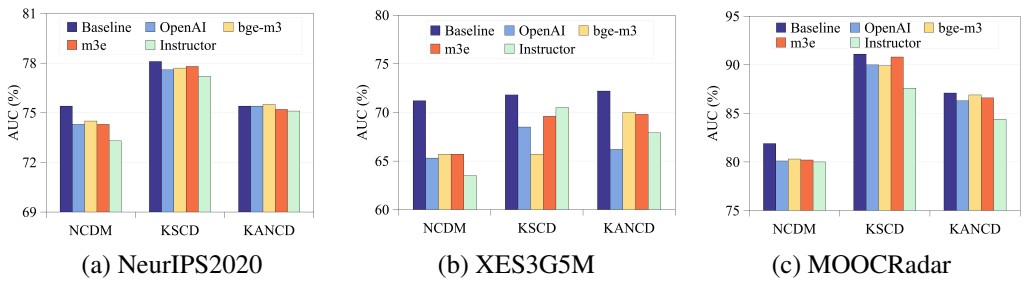

(a) NeurIPS2020          (b) XES3G5M          (c) MOOCRadar

Figure 4: Directly utilized text embedding may not benfit CDMs.

evaluation metric like previous works Wang et al. (2020a). We find that in all datasets, using the original ID-embedding performs better than almost all text embeddings. This further validates our conclusion.

**Details about right subfigure in Figure 1.** In this study, we first utilize the text-embedding-ada-002 to vectorized the exercise text from NeurIPS2020 dataset. We employ t-SNE Van der Maaten & Hinton (2008), a renowned dimensionality reduction method, to map the text embeddings onto a two-dimensional plane. Then, we use a scatter plot to visualize all the exercises in a two-dimensional space, coloring them by different concepts. To make the plot clear and understandable, we select the eight concepts that cover the most exercises as examples.

**The visualization of exercise text embeddings.** We employ t-SNE Van der Maaten & Hinton (2008) to map the exercise text semantic embeddings onto a two-dimensional plane. By shading the scatter plot according to the corresponding correct rates of exercise, with deeper shades of color indicating higher correct rates, we achieve a visual representation of the exercise' text feature distribution. The exercise text embeddings are relatively loose in the distribution of accuracy, which cause exercises

with high accuracy are not clustered together. This distribution will also lead to difficulties in using the exercise text embeddings later, which is also one of the reasons why we use exercise-refiner.

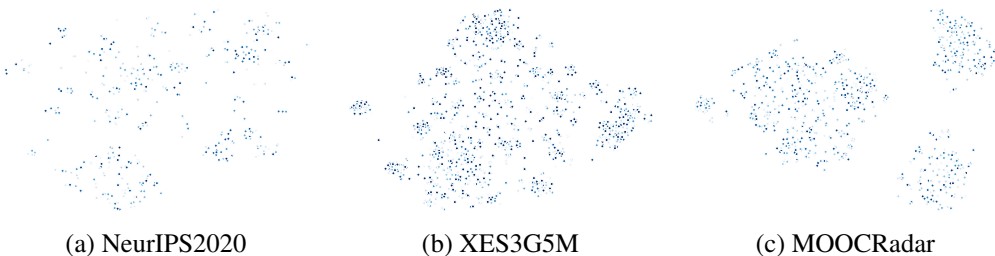

(a) NeurIPS2020  (b) XES3G5M  (c) MOOCRadar

Figure 5: The visualization of exercise text features.

## C  DETAILS ABOUT THE DFCD

### C.1  EXERCISE REFINER

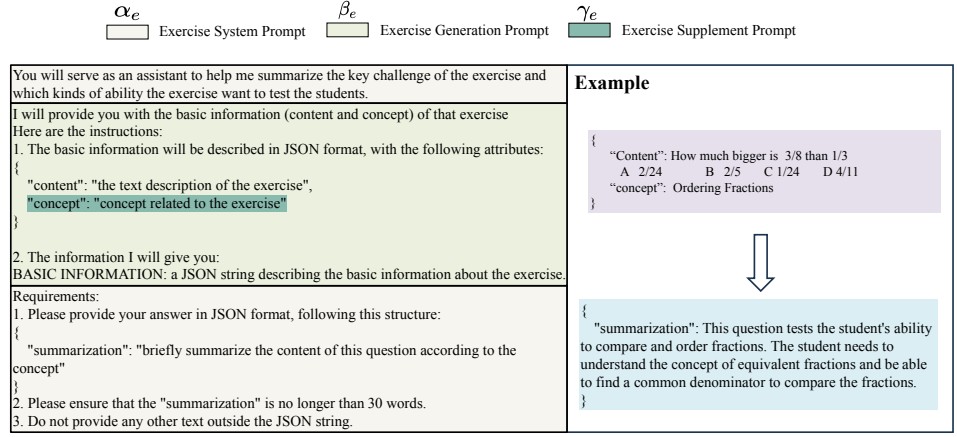

Figure 6: Prompt of refining exercises.

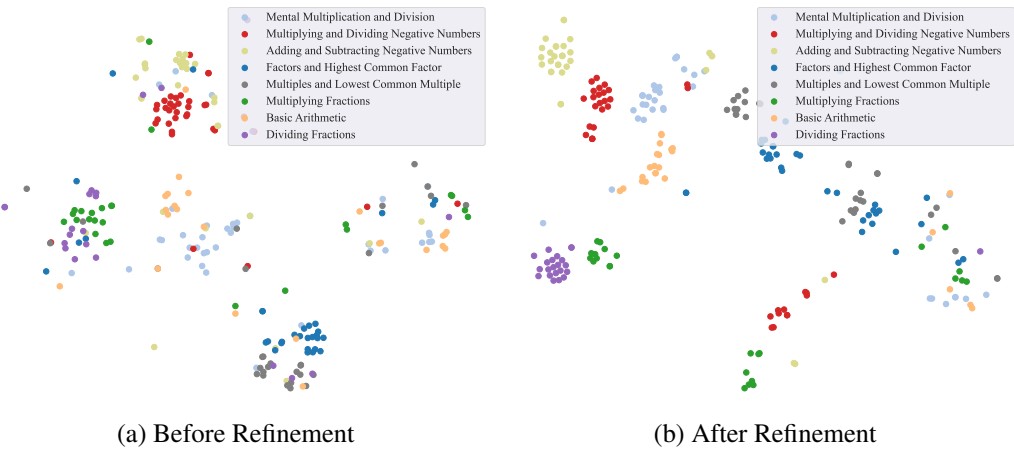

(a) Before Refinement  (b) After Refinement

Figure 7: t-SNE visualization comparing the exercise embeddings via OpenAI from the NeurIPS2020 dataset before and after refinement. (a) before refinement (b) after refinement.

Table 4: The quantitative metrics about clustering comparison with the knowledge concepts semantic feature before and after refinement. In each row, an entry with the best value is marked in bold.

| Metric | Before Refinement | After Refinement |
|---|---|---|
| Silhouette Score↑ | -0.3535 | **-0.2434** |
| Davies-Bouldin Index↓ | 17.7826 | **9.2424** |
| Calinski-Harabasz Index↑ | 7.4457 | **12.0039** |

Here, we first provide the detailed prompt of refining exercises in Figure 6. Detailed analysis can be found in Section 4.1. As shown in Figure 7, we can see that after refinement by the exercise refiner, exercises with the same concept are clustered more closely together, indicating that their representations better reflect the expert-labeled concepts. Moreover, we also provide detailed quantitative metrics in Table 4 about inter-cluster and intra-cluster distances comparison before and after the refinement to offer a more rigorous perspective. Following is brief introduction of our measurement.

• Silhouette Score: Measure the compactness of each point within its cluster and its separation from the nearest cluster. A value closer to 1 indicates better clustering performance.

• Davies-Bouldin Index: Measure the ratio of inter-cluster distance to intra-cluster distance. A smaller value indicates smaller intra-cluster distances and larger inter-cluster distances, signifying better clustering performance.

• Calinski-Harabasz Index: It calculates the ratio of intra-cluster variance to inter-cluster variance. A larger value indicates smaller intra-cluster variance and larger inter-cluster variance, signifying better clustering performance.

## C.2 CONCEPT REFINER

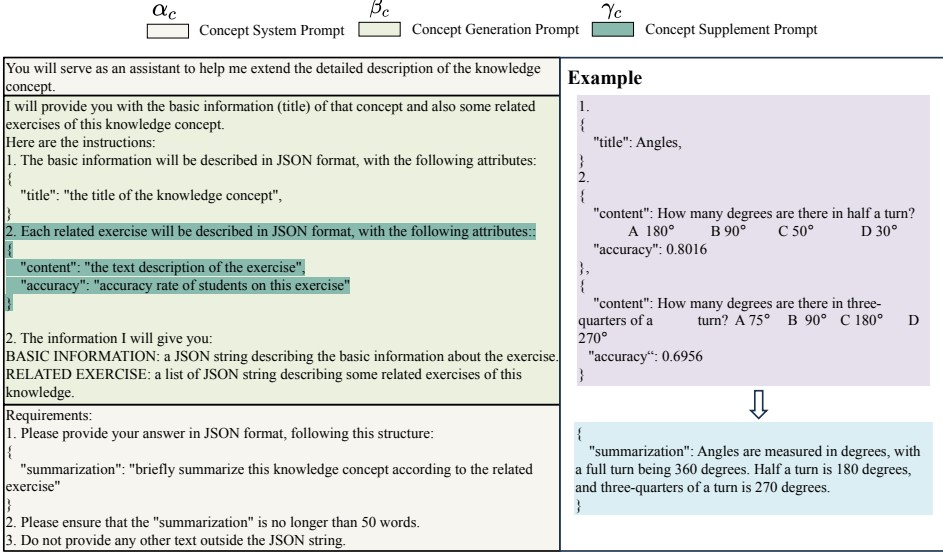

Figure 8: Prompt of refining concepts.

Here, we first provide the detailed prompt of refining concepts in Figure 8. Apparently, the concept name "Angles" may belong to multiple domains. However, through our designed prompt, we have successfully refined the concept of "Angles".

## C.3 POSITIVE MLP

In educational measurement Sympson (1978), the interaction function must meet the monotonicity assumption, meaning that more capable students should have higher accuracy rates. Akin to

NCDM Wang et al. (2020a), we employ MLP and use ReLU to ensure non-negative weights, thereby fulfilling the monotonicity assumption, referred to as Positive MLP.

## C.4 TIME COMPLEXITY OF DFCD

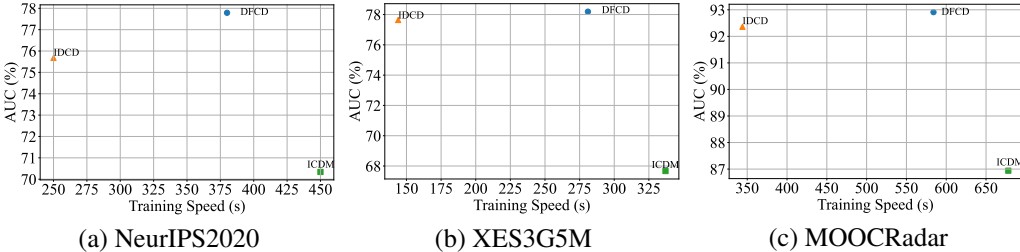

(a) NeurIPS2020        (b) XES3G5M        (c) MOOCRadar

Figure 9: Training speed comparision with IDCD and ICDM.

In this subsection, we present a detailed time complexity analysis of our proposed DFCD. For brevity, we do not include the time complexity of the integrated CDMs, as it can easily add to the overall time complexity of DFCD. Suppose that we have obtain the refined textual feature of students, exercises and concepts. We set the default graph encoder as GT. Firstly, we introduce some notions for clarity. $d$ is the latent dimension transformed after projectors. $L$ denotes the GT layer used in the graph encoder, $d_l$ denotes the dimension of textual features, and $F$ denotes the total number of students, exercises, and concepts. As the Textual-Projector and Response-Projector each have three MLPs, the total time complexity is $O(3d_l d + 3Fd)$. The time complexity of personalized attention module is $O(3Fd^2)$. The main time complexity of graph convolution is $O(LFd^2)$. So the ultimate time complexity of DFCD is $O(LFd^2 + 3d_l d + 3Fd + 3Fd^2)$. Therefore, the running speed of DFCD is related to the size of $Fd^2$, where $F$ depends on the nature of the dataset, and $d$ is a variable parameter. The smaller $d$ is, the slower the speed. In fact, as shown in Figure 9, our proposed DFCD has a faster training speed than ICDM, though it is slightly slower than IDCD. However, it achieves a higher AUC compared to both. **Notably, after training, we can infer the mastery level of 126 newly arrived students with 1,024 response logs in just 64 ms.**

## D EXPERIMENTAL DETAILS

### D.1 DATASETS INTRODUCTION AND DATA PREPROCESSING

• NeurIPS2020 Wang et al. (2020b): NeurIPS2020 comes from the public competition dataset of the NeurIPS 2020 Education Challenge. This competition mainly provides data on students' response logs to Eedi math problems in two school years (September 2018 to May 2020). Eedi provides diagnostic questions for students in elementary school through high school (approximately ages 7 to 18). Each diagnostic question is a multiple choice question with 4 possible answer choices, only one of which is correct. This competition mainly has 4 tasks. We choose the datasets of the 3rd and 4th tasks which include the English contextual information about the exercises and concepts, and the text information of the exercises does not exist in the datasets of tasks 1 and 2.

• XES3G5M Liu et al. (2024b): XES3G5M is a large-scale knowledge tracing benchmark dataset which consists of student interaction logs collected from a K-12 online learning platform in China. It contains rich auxiliary information about questions and their associated knowledge components. It contains the rich Chinese contextual information including tree structured KC relations, question types, textual contents and analysis.

• MOOCRadar Yu et al. (2023): MOOCRadar is a dataset for supporting the developments of cognitive student modeling in MOOCs. It provides the relevant learning resources, structures, and contents about the students' exercise behaviors. It also contains the Chinese contextual information about the exercises and concepts.

For the above datasets, we randomly selected 2,000 students in each dataset. This number is already a relatively large number for cognitive diagnosis tasks which can well support the training of the different cognitive diagnosis algorithms and evaluate their performance. At the same time, in order to

ensure that each selected student has enough exercise data to support his or her cognitive diagnosis, we only select students who answered more than 50 questions. It is worth noting that since the knowledge concepts of XES3G5M are displayed by tree structure, in order to avoid ambiguity, we only use the knowledge concepts of leaf nodes. We provide the three downloaded datasets, the result of refined text embeddings and the detailed code for data preprocessing in the anonymous repository at `https://anonymous.4open.science/r/DFCD-8710`.

### D.2 EVALUATION METRICS

**Classification Metrics.** Due to the unavailability of actual student mastery levels, we utilize inferred mastery levels by CDMs to predict student performance on exercises, as it is a binary classification problem (right or wrong). Following previous work, we use AUC and ACC as evaluation metrics.

**Degree of Agreement.** The underlying intuition here is that, if $s_a$ has a greater accuracy in answering exercises related to $c_k$ than student $s_b$, then the probability of $s_a$ mastering $c_k$ should be greater than that of $s_b$. Namely, $\mathbf{Mas}_{s_a,c_k} > \mathbf{Mas}_{s_b,c_k}$. DOA is defined as equation 11

$$\mathrm{DOA}_k = \frac{1}{Z} \sum_{a,b \in S} \delta\left(\mathbf{Mas}_{s_a,c_k}, \mathbf{Mas}_{s_b,c_k}\right) \frac{\sum_{j=1}^{M} \mathbf{Q}_{jk} \wedge \varphi(j,a,b) \wedge \delta(r_{aj},r_{bj})}{\sum_{j=1}^{M} \mathbf{Q}_{jk} \wedge \varphi(j,a,b) \wedge I(r_{aj} \neq r_{bj})}, \quad (11)$$

where $Z = \sum_{a,b \in S} \delta(\mathbf{Mas}_{s_a,c_k}, \mathbf{Mas}_{s_b,c_k})$, $\mathbf{Q}_{jk}$ indicates exercise $e_j$'s relevance to concept $c_k$, $\varphi(j,a,b)$ checks if both students $s_a$ and $s_b$ answered $e_j$, $r_{aj}$ represents the response of $s_a$ to $e_j$, and $I(r_{aj} \neq r_{bj})$ verifies if their responses are different, $\delta(r_{aj}, r_{bj})$ is 1 for a right response by $s_a$ and a wrong response by $s_b$, and 0 otherwise.

### D.3 EVALUATION IN DFCD

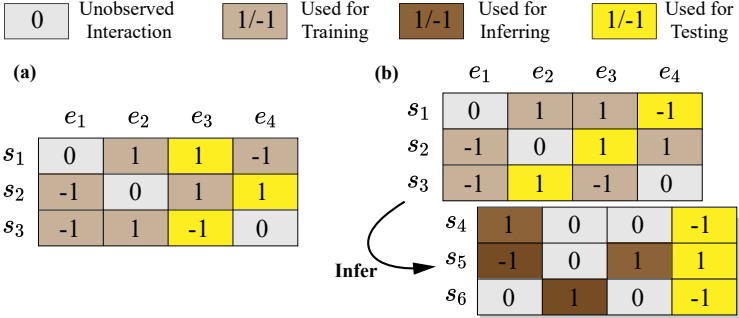

Figure 10: (a) Standard Scenario. (b) Unseen Student Scenario. we give example of Evaluation in unseen student scenario and the processes for other scenarios in open student learning environments are similar. For brevity and aesthetics, we omit the validation set.

Here, we provide an example about evaluation in DFCD of unseen student scenario, which is shown in Figure 10.

### D.4 SELECTION OF LLMS

In exercise-refiner and concept-refiner, we use OpenAI's large language model GPT-3.5-Turbo. Although OpenAI's GPT-4 has superior performance in terms of text generation quality, it is relatively expensive to use. Since the task of this paper is not that complicated, using GPT-3.5-Turbo can also achieve a relatively satisfactory result. The overall inference cost of GPT-3.5-Turbo in the task is about 3-4 US dollars, which is very cost-effective. At the same time, we also try Google's Gemini Pro Team et al. (2023). Although Gemini Pro is not as good as GPT-3.5-Turbo in terms of text generation quality, the performance on the task of this paper did not drop too much. And due to the free-use of Gemini Pro, it may also be a good choice.

Table 5: Overall prediction performance in standard scenario. Details are the same as Table 2.

| Datasets | NeurIPS2020 | | | XES3G5M | | | MOOCRadar | | |
|---|---|---|---|---|---|---|---|---|---|
| Metric | AUC | ACC | DOA@10 | AUC | ACC | DOA@10 | AUC | ACC | DOA@10 |
| MIRT | 77.79 | 70.72 | - | **79.47** | 83.45 | - | 92.52 | 91.23 | - |
| NCDM | 75.44 | 68.61 | 72.33 | 71.18 | 81.15 | 62.80 | 81.87 | 88.60 | 76.94 |
| RCD | 77.84 | 70.83 | 74.27 | 78.83 | 83.25 | 72.29 | OOM | OOM | OOM |
| KSCD | 78.07 | **71.23** | 58.53 | 71.80 | 81.75 | 57.92 | 91.05 | 87.92 | 49.79 |
| KANCD | 75.74 | 68.85 | 71.25 | 72.16 | 82.17 | 58.35 | 87.13 | 89.22 | 73.58 |
| DCD | 75.93 | 69.71 | 73.09 | 52.66 | 81.75 | 55.02 | 63.90 | 88.97 | 55.22 |
| IDCD | 77.33 | 70.24 | 74.27 | 76.28 | 82.60 | 70.40 | 92.18 | 91.28 | 80.93 |
| ICDM | 77.16 | 70.33 | 64.29 | 74.49 | 82.07 | 63.64 | 92.96 | 91.36 | 73.14 |
| **DFCD** | **78.11**$^*$ | 71.20 | **74.37** | 79.34 | **83.48** | **72.53** | **92.97** | **91.61**$^*$ | **81.01** |

## D.5 EXPERIMENT FOR STANDARD SCENARIO

**Baselines.** We conduct a comparison of DFCD against other baselines and utilize the hyperparameter settings described in their respective original publications. Among them, ICDM and IDCD can also be used in standard scenario, so we also add them in the baselines. As these two models has been introduced in the Section 5.2, introduction will not be given again. Due to the **Mas** inferred by MIRT being non-interpretable (i.e., the dimensions do not correspond to the number of concepts), we follow previous work Chen et al. (2023) by presenting MIRT results but not comparing them.

• MIRT Sympson (1978) is a representative model of latent factor CDMs, which uses multidimensional $\theta$ to model the latent abilities. We set the latent dimension as 16 which is the same as Wang et al. (2020a)

• NCDM Wang et al. (2020a) is a deep learning based CDM which uses MLPs to replace the traditional interaction function (i.e., logistic function).

• KaNCD Wang et al. (2023) improves NCDM by exploring the implicit association among knowledge concepts to address the problem of knowledge coverage.

• KSCD Ma et al. (2022) explores the implicit association among knowledge concepts and leverages a knowledge-enhanced interaction function.

• RCD Gao et al. (2021) leverages GNN to explore the relations among students, exercises and knowledge concepts. We utilize the student-exercise-concept component of RCD to construct the relation graph.

• DCD Chen et al. (2023) utilize students' response records to model student proficiency, exercise difficulty and exercise label distribution concepts.

**Details.** In line with prior CDM studies Wang et al. (2020a), in the standard scenario, we partition the data into train and test data and assess our model's performance on the test data. The test size is also set to 0.2, following the setting of the open student learning environment scenario. To ensure fairness in comparison, we adhere to the hyperparameter settings as specified in their original publications. Details can be found in Appendix D.6. MIRT are non-interpretable models, namely latent factor CDMs, the **Mas** it learns cannot be correlated directly with specific knowledge concepts. Therefore, it is not suitable for calculating DOA. In Table 5, we use "-" to indicate this inapplicability. If CDMs signify out-of-memory on an NVIDIA 3090 GPU, we use the term "OOM" to denote this occurrence.

**Results.** The comparison results are listed in Table 5. As we can see, despite DFCD is primarily tailored for the open student learning environment scenario in CD, it performs competitively with or even outperforms most of the current state-of the-art CDMs in predictive performance. Moreover, DFCD demonstrates commendable interpretability performance across all three datasets.

## D.6 IMPLEMENTATION AND BASELINES' DETAILS

This section delineates the detailed settings when comparing our method with the baselines and state-of-the-art methods in both standard scenario and open student learning environments. All

experiments are run on a Linux server with two 3.00GHz Intel Xeon Gold 6354 CPUs and one RTX3090 GPU. All the models are implemented by PyTorch Paszke et al. (2019). For all methods that involve using Positive MLP as the interaction function, we adopt the commonly used two-layer tower structure with hidden dimensions of 512 and 256.

In the following, we elaborate on some details regarding the utilization of compared methods.

**Baselines in Standard Scenario.**

• MIRT Sympson (1978) is a representative model of latent factor CDMs, which uses multidimensional $\theta$ to model the latent abilities. We set the latent dimension as 16 which is the same as Wang et al. (2020a)

• NCDM Wang et al. (2020a) is a deep learning based CDM which uses MLPs to replace the traditional interaction function (i.e., logistic function). We adopt the default parameters which are reported in that paper.

• RCD Gao et al. (2021) leverages GNN to explore the relations among students, exercises and knowledge concepts. Here, to ensure a fair comparison, we solely utilize the student-exercise-concept component of RCD, excluding the dependency on concepts.

• KaNCD Wang et al. (2023) improves NCDM by exploring the implicit association among knowledge concepts to address the problem of knowledge coverage. Here, we adopt the default parameters reported in that paper. For instance, the latent dimension is set to 20, and the default type is selected as GMF.

• KSCD Ma et al. (2022) also explores the implicit association among knowledge concepts and leverages a knowledge-enhanced interaction function. Here, we adopt the default parameters reported in that paper. The latent dimension is set to 20, and the default interaction function utilizes its proposed one on NeurIPS2020 and XES3G5M. We set the interaction function to NCDM because KSCD encounters out-of-memory issue on MOOC-Radar.

**Baselines in Open Student Learning Environments.**

• IDCD Li et al. (2024): It propose an identifiable cognitive diagnosis framework based on a novel response-proficiency response paradigm and its diagnostic module leverages inductive learning representations which can be used in the open student learning environment.

• ICDM Liu et al. (2024a): It utilizes a student-centered graph and inductive mastery levels as the aggregated outcomes of students' neighbors in student-centered graph which enables to infer the unseen students by finding the most suitable representations for different node types.

The implementation of MIRT, NCDM and KaNCD comes from the public repository `https://github.com/bigdata-ustc/EduCDM`. For RCD, IDCD, ICDM and KSCD, we adopt the implementation from the authors in `https://github.com/bigdata-ustc/RCD`, `https://github.com/CSLiJT/ID-CDF`, `https://github.com/ECNU-ILOG/ICDM` and `https://github.com/BIMK/Intelligent-Education/tree/main/KSCD_Code_F`.

## D.7 ABLATION STUDY

Here, we provide the complete result of the ablation study in Table 6. The analysis can be found in Section 5.2.

## D.8 GENERALIZATION ANALYSIS

Here, we provide the complete result of generalization experiment in Figure 11 and Table 7. The analysis can be found in Section 5.2. Generalization analysis indicates that even in environments where data is sparse or not well-structured, the model's performance remains robust, thereby expanding its applicability. This generalization allows the model to perform well across a wide range of conditions, making it versatile and suitable for various educational contexts, including those where data may be incomplete or inconsistent.

Table 6: Overall prediction performance of ablation study for DFCD in open student learning environment scenario. Details are as same as Table 2.

| Dataset | | NeurIPS2020 | | | XES3G5M | | | MOOCRadar | |
|---|---|---|---|---|---|---|---|---|---|
| Metric | AUC | ACC | DOA@10 | AUC | ACC | DOA@10 | AUC | ACC | DOA@10 |
| | | | | Unseen Student | | | | | |
| DFCD-w.o.TE | 78.02 | 71.28 | 74.23 | 77.78 | 83.12 | 72.20 | 92.67 | 91.53 | 82.24 |
| DFCD-w.o.RE | 78.12 | 71.08 | 74.14 | 77.72 | 83.04 | 72.07 | 92.90 | 91.35 | **82.64** |
| DFCD-w.o.attn | 78.11 | 71.31 | 74.26 | 77.80 | 83.10 | 72.17 | 92.90 | 91.60 | 81.32 |
| DFCD | **78.19** | **71.39** | **74.33** | **77.81** | **83.18** | **72.21** | **92.91** | **91.68** | 82.15 |
| | | | | Unseen Exercise | | | | | |
| DFCD-w.o.TE | 77.72 | 71.14 | 74.13 | 75.90 | 82.41 | 72.06 | 91.97 | 91.52 | 82.02 |
| DFCD-w.o.RE | 74.59 | 68.38 | 74.11 | 68.21 | 81.06 | 71.71 | 85.94 | 89.16 | **82.37** |
| DFCD-w.o.attn | 77.74 | 71.27 | 74.10 | 76.10 | 82.56 | 71.91 | 91.92 | 91.51 | 81.96 |
| DFCD | **77.76** | **71.31** | **74.17** | **76.11** | **82.62** | **72.29** | **91.98** | **91.61** | 81.93 |
| | | | | Unseen Concept | | | | | |
| DFCD-w.o.TE | 77.67 | 70.80 | 74.07 | 78.82 | 83.38 | 72.03 | 92.55 | 91.33 | 82.34 |
| DFCD-w.o.RE | 76.80 | 69.72 | 74.13 | 76.83 | 82.45 | 72.03 | 91.84 | 90.76 | **82.67** |
| DFCD-w.o.attn | 77.63 | 70.63 | 73.85 | 78.46 | 83.30 | 72.02 | 92.88 | 91.50 | 80.81 |
| DFCD | **77.68** | **70.83** | **74.14** | **78.83** | **83.41** | **72.14** | **92.89** | **91.56** | 80.56 |

Table 7: The performance comparison with DFCD and BetaCD in cold-start scenario where new student response logs are sparse. Size means the size of response logs per new student.

| Datasets | | NeurIPS2020 | | XES3G5M | |
|---|---|---|---|---|---|
| Metric | Size | BetaCD | DFCD | BetaCD | DFCD |
| AUC | 3 | 69.17 | 68.42 | 72.05 | 71.43 |
| | 5 | 69.71 | 68.81 | 72.64 | 72.01 |
| | 10 | 71.23 | 71.46 | 73.25 | 73.22 |
| ACC | 3 | 64.14 | 63.53 | 82.40 | 81.53 |
| | 5 | 64.56 | 64.53 | 82.47 | 81.54 |
| | 10 | 65.13 | 65.81 | 82.41 | 81.78 |
| RMSE | 3 | 46.95 | 47.21 | 36.47 | 37.16 |
| | 5 | 46.80 | 46.84 | 36.38 | 37.09 |
| | 10 | 46.36 | 46.26 | 36.28 | 36.85 |

### D.9 HYPERPARAMETER ANALYSIS

Here, we provide the complete result of hyperparameter experiment in Figure 12. The analysis can be found in Section 5.3.

### D.10 TEXT EMBEDDING ANALYSIS

To demonstrate the impact of different text embedding models on DFCD across different datasets and scenarios, we select four competitive text embedding models currently available:

• Text-embedding-ada-002 Brown et al. (2020): As OpenAI's leading text embedding model, it outperforms most embedding models in tasks such as text search, code search, and sentence similarity. It is widely recognized as one of the best text embedding model available today.

• BGE-M3 Chen et al. (2024): A multi-lingual, multi-functionality, multi-granularity text embedding model through self-knowledge distillation. It can support more than 100 working languages, leading to new state-of-the-art performances on multi-lingual and cross-lingual retrieval tasks.

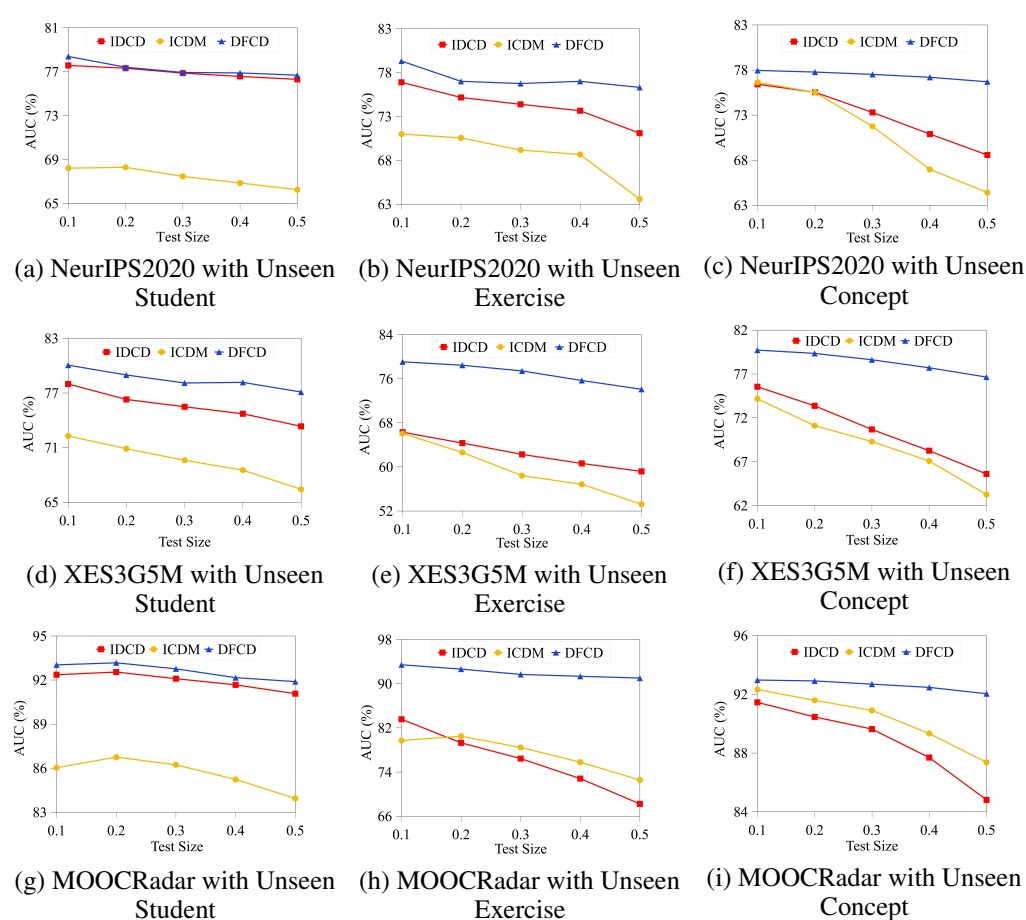

Figure 11: Comparison with other CDMs in different test sizes.

• M3E-base Wang Yuxin (2023): This open-source model is evaluated on a large-scale sentence-pair dataset that includes 22 million samples across domains such as Chinese Wikipedia, finance, healthcare, law, news, and academia. M3e-base is primarily designed for Chinese contexts, making it suitable for the XES3G5M and MOOCRadar datasets, which include Chinese exercise text.

• Instructor-base Su et al. (2022): This model introduces INSTRUCTOR, a novel method for computing text embeddings based on task instructions. It generates text embeddings tailored to various downstream tasks and domains without further training, aligning with our application's requirements.

As shown in Figure 13, in most scenarios and datasets, text-embedding-ada-002 and bge-m3 demonstrate superior performance, likely due to their extensive training data, which supports them to better captures semantic information. Their versatility across multiple languages and functions makes them effective for both English exercise text in the NeurIPS2020 dataset and Chinese exercise text in the XES3G5M and MOOCRadar datasets. M3e-base, being primarily suited for Chinese contexts, performs well on the Chinese exercise text in XES3G5M and MOOCRadar datasets but shows weaker performance on the English exercise text in NeurIPS2020 dataset. The instructor-base model, which relies heavily on instruction guidance, may only perform well in specific scenarios. While it is possible that a more suitable instruction could improve its performance in the specific scenarios, but this falls outside the scope of our study and will not be further discussed.

## D.11 DIAGNOSIS RESULT ANALYSIS

Here, we provide the complete result of visualization of diagnosis result in Figure 14. Indeed, students can naturally be grouped into categories based on their scores, such as those with low and high correct

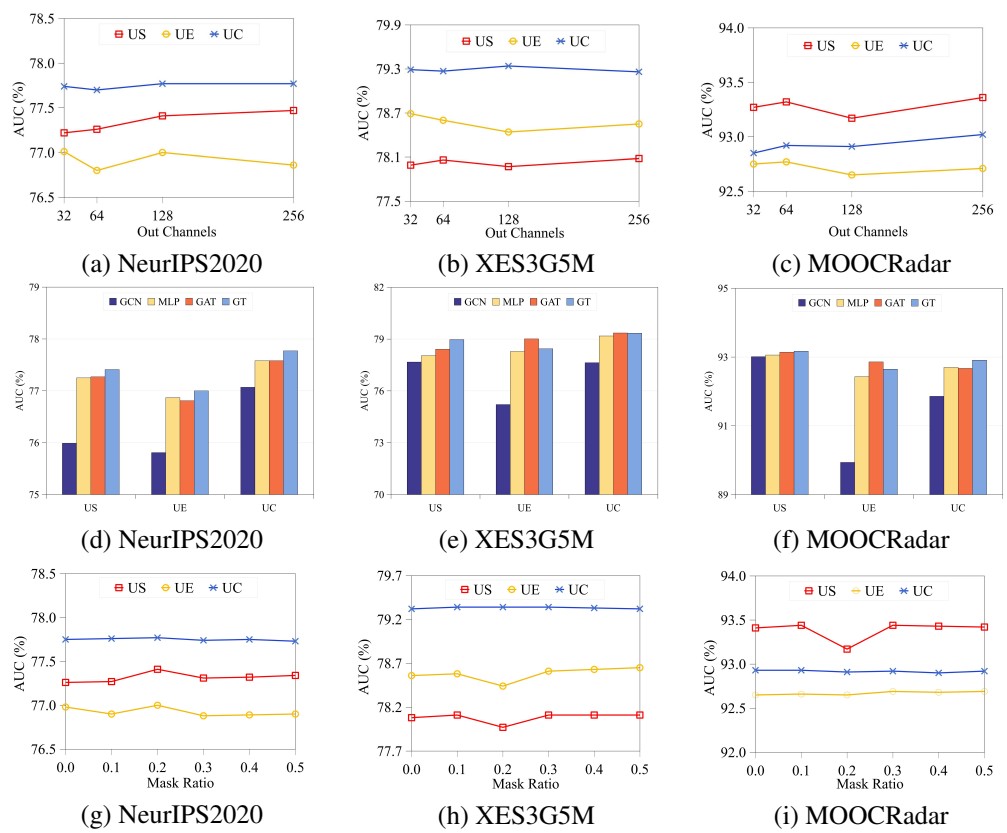

Figure 12: Comparison of DFCD with different hyperparameters. US means the scenario of unseen student, UE means the scenario of unseen exercise, UC means the scenario of unseen concept.

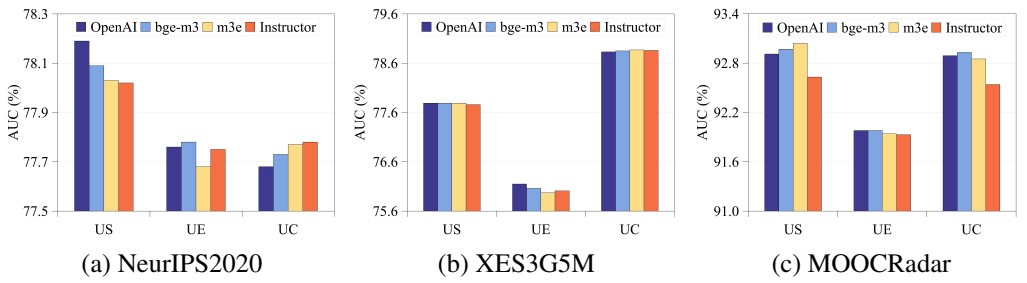

Figure 13: Comparison of DFCD with different text embedding module. US means the scenario of unseen student, UE means the scenario of unseen exercise, UC means the scenario of unseen concept.

rates. This classification reflects intrinsic differences in their mastery levels. Details can be found in Appendix D.11. We employ t-SNE Van der Maaten & Hinton (2008), a renowned dimensionality reduction method, to map the inferred **Mas** by CDMs onto a two-dimensional plane. By shading the scatter plot according to the corresponding correct rates, with deeper shades of color indicating higher correct rates, we achieve a visual representation of the students' **Mas** distribution. Notably, historical students are colored in blue, while newly arrived students are colored in green. We compare our DFCD with IDCD in three different open student learning environment scenarios. As shown in Figure 14, DFCD displays a long strip trend, with the color of the points on the strip gradually changing from lighter to darker shades. This indicates that DFCD successfully captures both the historical and new students' **Mas** trends. In contrast, the color distribution of IDCD is relatively loose, suggesting it may fail to accurately capture students' **Mas** information. Moreover, the mastery levels of new students inferred by DFCD are more reliable, as new students with similar correct rates (colored in green) cluster closely with historical students (colored in blue) of comparable rates.

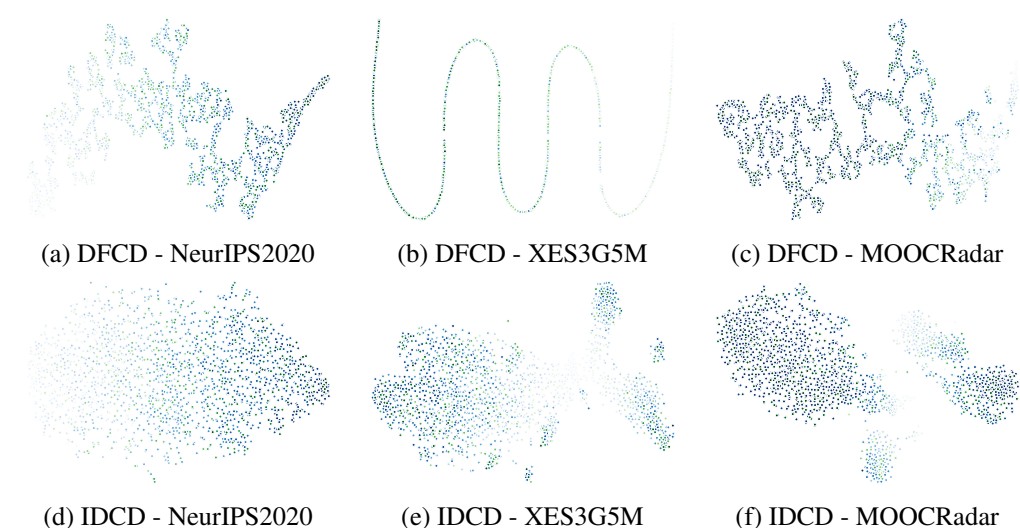

(a) DFCD - NeurIPS2020    (b) DFCD - XES3G5M    (c) DFCD - MOOCRadar

(d) IDCD - NeurIPS2020    (e) IDCD - XES3G5M    (f) IDCD - MOOCRadar

Figure 14: t-SNE scatter plots for DFCD and IDCD on the NeurIPS2020 dataset. Blue color is used to mark observed students, exercises, and concepts, while green is used to mark unobserved students, exercises, and concepts. The intensity of the color represents the correct rate.

