# OpenReview forum: "A Dual-Fusion Cognitive Diagnosis Framework for Open Student Learning Environments"
_ICLR.cc/2025/Conference — ICLR 2025 Conference Withdrawn Submission_

### Official Review · Reviewer_45Wj · 2024-10-29

**Soundness:** 1
**Presentation:** 2
**Contribution:** 1
**Rating:** 3
**Confidence:** 5

**Summary:**

This paper proposes a new cognitive diagnosis model that incorporates the textual semantic of the exercises in the modeling process. For this, the proposed work first refines the exercises and concepts via LLMs, and leverages these embeddings in a fusion network to merge with the response features. The proposed model is trained for an accurate prediction of student performances. The experiments are performed in three online learning datasets.

**Strengths:**

- The paper is easy to follow, except for some specific sections.
- This work incorporates the textual semantic into CDM.
- The code repository is structured well.

**Weaknesses:**

My comments are as below:

- Lines 79-83:I strongly disagree with the authors’ statements there. First of all, it is not the textual features that generalize over the unseen domains, but the large language models. Otherwise, one could simply train any classical (i.e. pre-LLM) language model over the same textual features and there would be no need for further research. Secondly, even the most recent LLMs have various issues in generalization and that is why many researchers are still working on instruction-finetuning, long-context memorization, reasoning, etc, just to name a few. Therefore, the next sentence from the authors “All we need is to train a projector to map the textual space to the actual diagnostic space” is extremely simplistic and not supported.

- Details are missing for motivation study. Authors are showing a counter-intuitive result in the Figure 1 and Appendix B that the simply randomly initialized embeddings perform better than the text embeddings acquired from LLMs. Particularly, as can be seen from the right subfigure of Figure 1, the embeddings are in fact much more meaningful then any standard random initialization of ID embeddings. Therefore, it is not convincing to see that somehow these neural CD models perform worse with more meaningful embeddings. This requires more explanation and also the details about how these experiments are conducted in the first place. As final note, if the randomly initialized (i.e. ID) embeddings are trained and the text embeddings are kept fixed during the experiment, it is not a fair comparison. 1) It means that the baselines have more degree of freedom (i.e. number of parameters) than the version with text embeddings, which is an unfair advantage and 2) fixed embeddings should be subject to training for a fair comparison.

- In Related Work, the authors claimed TechCD and ZeroCD are different from their focus. If these works are also centered on exercise texts, I could not follow what makes them different in terms of focus. I think this part should be communicated better.
Lines 199-202: The authors have the following claim “The exercise text can, to some extent, reflect the difficulty level of specific concepts for the students. However, it is evident that exercise text alone cannot directly reflect the expert annotated concepts being tested.“, which requires further verification. First of all, it is not clear how the exercise text can reflect the difficulty level. Second, it is not clear why the exercise text cannot reflect the associated concept. Even if it doesn’t match the annotated concept completely, it could be attributed to the domain expert (as shown in [1]) but not to the model directly. Further, the authors’ motivating example about “Square Roots” cannot be found in their Figure 2 at all, it makes it extra confusing.

- (Section 4.2) The design of Response Feature Constructor looks like an arbitrary choice and it is also poorly explained. First of all, the authors criticize the usage of historical interaction matrix $I^O$ but they also leverage the exact same matrix. As a side note, $I^O$ is not formulated in the paper at all, therefore, it is hard to follow (here I am referring to the entries of the matrix, not the dimensionality, which can be inferred from num students and exercises).Further, this module also incorporate another matrix $Q^O$ and it is not introduced anywhere, so there is no way to know what it stands for. Further the choice of having some arbitrary 0-matrices inside the big $R^O$ is not communicated. For instance, why is student and concept vectors (within $R^0$) start and end with 0 vectors, but the opposite for exercise vectors? Overall, it is hard to find the motivation for such a design.

- (Section 4.3) The authors present dimensionality mismatch between two modalities as a key challenge in the formulation, although the solution is simple projection and attention which are applied in numerous cases over the last +5 years at least.

- (Section 4.3) Graph Encoder seems like an integral part of the framework, as the final prediction is directly leveraging the outputs from the graph encodings. Yet, it is surprising to see that almost no information is provided about how the graph is constructed and details of the function signature for their “Encoder(.)” function. If there are 3 entity types, i.e., students, exercises and concepts, how are they connected in the entire graph? As a follow-up, even more importantly, how are the “unseen” students, exercises and concepts added to the existing graph after the training? If they are not added after the training but instead added in the initialization phase, how can authors claim that those entities are “unseen”? Therefore, this part requires complete re-writing and extensive clarifications.

- Prediction task itself is unclear. Are the authors predicting only the next exercises’ performance for each student, or all of them at once? If they predict all the next exercises, then what is the number of exercises to be predicted?

- The authors use only a tiny portion of the original datasets and yet claim that it is a large number for the cognitive diagnosis task. For instance, original XES3G5M dataset [2] has +18K students (10 times of what is presented in the paper), 865 Knowledge Concepts (4 times of what is presented in the paper), 7652 exercises (5 times of what is presented in the paper) and +5.5M response logs (27 times of what is presented in the paper). Further, they claim avg. number of concepts per question is 1, which is also wrong. As noted by the authors of XES3G5M, it is 1.164 . Therefore, the entire study is conducted in a highly limited setup. This, of course, raises many important concerns about the scalability of the proposed approach. Considering that the proposed work needs to create a large matrix whose size is quadratically scaled up by the number of students, exercises and knowledge concepts, I think the entire dataset would not fit into the proposed work to be processed. This would imply that the authors’ approach cannot be scaled to real-world online education, which contradicts with their main claims in the paper.

- I am surprised to see that the term “Knowledge Tracing” (KT) is not mentioned in the main paper (except one place in related work), although the whole line of work is directly related to the current model. In fact, the datasets that the authors use are knowledge tracing datasets which have been benchmarked many times in the literature. Further, the performance prediction task is the exact task that is performed in knowledge tracing. That is why, the authors should have included KT models and compared their framework against those KT models in the experiments. In fact, even the first deep learning based KT model from 2015, DKT [3], can perform both the student performance prediction and output the knowledge mastery. Even this DKT model seems to outperform the performance of the proposed work in XES3G5M, as such an experiment rigorously performed by the authors there [2]. Of note, DKT can process the entire dataset unlike the proposed approach. Of course, DKT is not the only work that performs the same task as the authors. Some related models (among many others) include qDKT [4], IEKT [5], QIKT [6], DKVMN [7], DeepIRT [8], simpleKT [9], sparseKT [10] and many others. Therefore, I think this work is completely missing a crucial line of work.

- Code repository: In the paper, authors claim to provide the result of refined text embeddings. Then in the repository, they add a note that uploading the embeddings is not possible. I think this is completely misleading the readers as the promise is not delivered. As I completely understand the problem with uploading the raw embeddings, I believe the incorrect statement in the paper should be fixed.

Minors:

- Figure1 can be larger for the ease of readability. Especially the left and right subfigures.

- Line 201-202 typo: “For instance, as shown in Figure 2(a), it may **be** related to” .

[1] Steven Moore, Robin Schmucker, Tom Mitchell, and John Stamper. Automated generation and tagging of knowledge components from multiple-choice questions. In Learning@ Scale, 2024

[2] Zitao Liu, Qiongqiong Liu, Teng Guo, Jiahao Chen, Shuyan Huang, Xiangyu Zhao, Jiliang Tang, Weiqi Luo, and Jian Weng. XES3G5M: A knowledge tracing benchmark dataset with auxiliary information. In NeurIPS, 2023

[3] Chris Piech, Jonathan Bassen, Jonathan Huang, Surya Ganguli, Mehran Sahami, Leonidas J Guibas, and Jascha Sohl-Dickstein. Deep knowledge tracing. In NeurIPS, 2015.

[4] Shashank Sonkar, Andrew E Waters, Andrew S Lan, Phillip J Grimaldi, and Richard G Baraniuk. qDKT: Question-centric deep knowledge tracing. In EDM, 2020.

[5] Ting Long, Yunfei Liu, Jian Shen, Weinan Zhang, and Yong Yu. Tracing knowledge state with individual cognition and acquisition estimation. In SIGIR, 2021

[6] Jiahao Chen, Zitao Liu, Shuyan Huang, Qiongqiong Liu, and Weiqi Luo. Improving interpretability of deep sequential knowledge tracing models with question-centric cognitive representations. In AAAI, 2023.

[7] Jiani Zhang, Xingjian Shi, Irwin King, and Dit-Yan Yeung. Dynamic key-value memory networks for knowledge tracing. In WWW, 2017.

[8] Deep-IRT: Make deep learning based knowledge tracing explainable using item response theory. EDM, 2019.

[9] Zitao Liu, Qiongqiong Liu, Jiahao Chen, Shuyan Huang, and Weiqi Luo. simpleKT: A simple but tough-to-beat baseline for knowledge tracing. In ICLR, 2023.

[10] Shuyan Huang, Zitao Liu, Xiangyu Zhao, Weiqi Luo, and Jian Weng. Towards robust knowledge tracing models via k-sparse attention. In SIGIR, 2023.

**Questions:**

On top of my questions above, I would like to ask the following:

- As the exercises are solved in a specific order, each student history in fact generates a temporal sequence. Where is the temporality encoded in the authors’ work? If it is not encoded, then as a follow-up, how can one use the authors’ work to track the mastery level of students **over time** ?

---

### Official Review · Reviewer_wwHp · 2024-10-31

**Soundness:** 3
**Presentation:** 3
**Contribution:** 2
**Rating:** 3
**Confidence:** 4

**Summary:**

The paper titled "A Dual-Fusion Cognitive Diagnosis Framework for Open Student Learning Environments" introduces a novel framework (DFCD) that aims to enhance cognitive diagnosis models by integrating textual semantic features and response-relevant features. The authors propose the use of large language models as refiners for exercises and concepts to improve coherence and then employ text embedding models to capture semantic information. A dual-fusion module merges these features to enable inference in open student learning environments without retraining. The paper claims superior performance and adaptability in real-world datasets compared to existing CDMs.

**Strengths:**

1.	This paper tackles a significant challenge in intelligent education by proposing a framework capable of effectively operating in open student learning environments without the need for retraining. Notably, the integration of textual semantic features with response-relevant features through a dual-fusion module is innovative and addresses the limitations of current CDMs. Furthermore, employing large language models as refiners to enhance textual features represents a novel contribution to the field.

2.	The paper is technically rigorous, featuring extensive experiments across real-world datasets that demonstrate the effectiveness of the DFCD framework. The writing is clear, and the results are presented in an accessible manner, making them easy to follow and understand.

**Weaknesses:**

1.	The DFCD approach omits the modeling of exercise discrimination, a crucial aspect emphasized by traditional CDMs for accurately assessing a student’ proficiency.

2.	The rationale behind utilizing the exercise refiner, as illustrated in Figure 5, is not clearly defined. It raises the question of why exercises with similar accuracy should exhibit similar representations. Shouldn’t the representation of an exercise be more influenced by its associated knowledge points and difficulty level?

3.	While SimpleCD is characterized as parameter-free except for the interaction function, this assertion appears questionable due to its significant dependence on preceding modules, such as projection layers, attention models, and graph encoders.

4.	In Tables 2 and 5, the term DFCD is ambiguous, as it is unclear whether it refers to SimpleCD or the integration strategy discussed in Section 4.4. Providing a clearer definition of DFCD and explicitly distinguishing it from SimpleCD would improve the tables' comprehensibility and enhance the overall presentation.

**Questions:**

1.	In Section 4.2, what is the rationale for constructing R^{O}? Please elaborate on the statement regarding the imbalance in the size of the student and exercise feature spaces. Previous CDMs typically represent students and exercises as 1×|C|, which correlates with the number of knowledge points and reflects students' mastery of those points and the relationship between exercises and knowledge.

2.	I'm a bit curious, in scenarios involving unseen concepts, can DFCD diagnosis students' proficiency in these unseen concepts？

3.	The choice of datasets is unusual; why were the last two datasets selected given their high average correctness? Specifically, for the MOOCRadar dataset, even if the CDM predicts all responces as correct, it may still reflect a favorable accuracy. Are there other datasets that include text modalities but do not have such high average correctness? I believe experiments could be better expanded on more balanced datasets instead of relying on the high average correctness of MOOCRadar.

4.	Are there any experimental results related to the integration strategy discussed in Section 4.4?

5.	In Figure 10, what is the distinction between "infer" and "test"? Additionally, the color scheme for "s4-e1" appears inconsistent; it is advisable to standardize the color coding for clarity.

---

### Official Review · Reviewer_UWEi · 2024-11-03

**Soundness:** 2
**Presentation:** 1
**Contribution:** 2
**Rating:** 1
**Confidence:** 4

**Summary:**

This paper proposes a DFCD framework to address the challenge of aligning textual semantic and response-relevant features in open student learning environments. It includes a textual feature constructor using large language models and text embedding models, a response feature constructor with a novel response matrix, and a dual-fusion module. Experiments on real-world datasets show the effectiveness of this method.

**Strengths:**

1. The issues concerned by the authors are very important for intelligent education.
2. It is very important to pay attention to textual information in cognitive diagnosis tasks.

**Weaknesses:**

1. This article is very difficult to read, especially for readers who are not familiar with CD. For example, in line 42 of the introduction, what is the Q - matrix? It is hard to understand the meaning of the Q - matrix from Figure 1. Readers can only understand the meaning of the Q - matrix after reading section 3. In line 503, almost all experimental results are presented in the appendix. If authors believe that the 10 - page limit stipulated by ICLR is insufficient to present the experimental results, they can consider submitting to some journals, such as TPAMI.
2. In the textual feature constructor, the authors propose to use LLMs to summarize the exercise text. Has authors ever tried to directly send these texts into the text embedding (i.e., without summarizing through the LLM)?
3. The innovation of the author is limited. The idea of using text semantics to enhance the generalization ability of the model is very common in various fields, especially when LLM has demonstrated very strong generalization ability. I am not sure whether the method proposed by authors can inspire the work of researchers in other fields. Can authors list your contribution points?
4. The paper lacks a theoretical analysis of why the proposed method is effective.
5. The authors have compared relatively few experimental baselines. The authors explain the differences between DFCD and TechCD and ZeroCD in line 135. As far as I understand, TechCD and ZeroCD focus on the cold-start problem of students (new students), and DFCD can also solve the new students problem. Then why don't the authors compare TechCD and ZeroCD as baselines? Moreover, as methods focusing on cold-start, TechCD and ZeroCD should be introduced in detail by the authors instead of simply stating "unlike the cold-start issues addressed by TechCD and ZeroCD". Besides, why didn't the authors compare DCD and ECD mentioned in the related work (excluding the standard scenario)? Why only compare IDCD and ICDM in line 132?

**Questions:**

Please see  weaknesses for details.

---

### Official Review · Reviewer_yFDp · 2024-11-04

**Soundness:** 3
**Presentation:** 3
**Contribution:** 3
**Rating:** 6
**Confidence:** 3

**Summary:**

This paper proposes DFCD, a novel cognitive diagnosis method that integrate textual semantic features and response relevant features for open student learning environments, which can be generalized to unseen students, exercises, and concepts. The authors conduct comprehensive experiments across various datasets to demonstrate the superior performance of DFCD. This method can be potentially useful for researchers in the field.

**Strengths:**

1. The paper addresses an important problem in Cognitive Diagnosis: open student learning environment diagnosis, which is important as the online learning paradigm continues to evolve.
2. The method proposed by this paper is both sound and effective.
3. The paper is clearly written, with appropriate figures, tables, and equations.
4. The experiments design is thorough and extensive, providing comprehensive evidence for the claims made

**Weaknesses:**

This paper is clearly written with abundant experiments. I only have questions with a few details.
1. The preliminary experiment mentioned in the introduction and Appendix B shows that directly replacing id-embedding with textual embedding fails to benefit CDMs. Can this experiment be considered as an ablation study that removes the exercise refiner in the DFCD framework?
2. The performance of exercise refiner is only evaluated by how these exercises are clustered, but not how they benefit the final prediction of CD models.
2. Some typos: Line 136, Attribue -> Attribute; Line 310, Postive -> Positive;

**Questions:**

The exercise refiner shown in Figure 6 aims at providing textual information that is useful for CD (e.g.: what kind of skills students should have to complete the exercise). However, the summarization could omit detailed information for specific exercises. Two different exercises could have the same refined results.

---

### Note · Authors · 2024-12-03

I have read and agree with the venue's withdrawal policy on behalf of myself and my co-authors.